# Assessment of the Performances of the Protein Modeling Techniques Participating in CASP15 Using a Structure-Based Functional Site Prediction Approach: ResiRole

**DOI:** 10.3390/bioengineering10121377

**Published:** 2023-11-30

**Authors:** Geoffrey J. Huang, Thomas K. Parry, William A. McLaughlin

**Affiliations:** Department of Medical Education, Geisinger Commonwealth School of Medicine, 525 Pine Street, Scranton, PA 18509, USAtkparry1@geisinger.edu (T.K.P.)

**Keywords:** protein structure prediction, protein model quality assessment, functional site prediction, benchmarking, CASP15 assessment

## Abstract

Background: Model quality assessments via computational methods which entail comparisons of the modeled structures to the experimentally determined structures are essential in the field of protein structure prediction. The assessments provide means to benchmark the accuracies of the modeling techniques and to aid with their development. We previously described the ResiRole method to gauge model quality principally based on the preservation of the structural characteristics described in SeqFEATURE functional site prediction models. Methods: We apply ResiRole to benchmark modeling group performances in the Critical Assessment of Structure Prediction experiment, round 15. To gauge model quality, a normalized Predicted Functional site Similarity Score (PFSS) was calculated as the average of one minus the absolute values of the differences of the functional site prediction probabilities, as found for the experimental structures versus those found at the corresponding sites in the structure models. Results: The average PFSS per modeling group (gPFSS) correlates with standard quality metrics, and can effectively be used to rank the accuracies of the groups. For the free modeling (FM) category, correlation coefficients of the Local Distance Difference Test (LDDT) and Global Distance Test-Total Score (GDT-TS) metrics with gPFSS were 0.98239 and 0.87691, respectively. An example finding for a specific group is that the gPFSS for EMBER3D was higher than expected based on the predictive relationship between gPFSS and LDDT. We infer the result is due to the use of constraints imprinted by function that are a part of the EMBER3D methodology. Also, we find functional site predictions that may guide further functional characterizations of the respective proteins. Conclusion: The gPFSS metric provides an effective means to assess and rank the performances of the structure prediction techniques according to their abilities to accurately recount the structural features at predicted functional sites.

## 1. Introduction

Protein structure prediction via computational methods requires accurate ways to assess the quality of the structure models produced. Upon the availability of a protein structure determined using an experimental technique, such as X-ray crystallography, the structure prediction techniques used to generate the protein models prior to that time can be benchmarked by directly comparing the models with the experimental structure. Example metrics that compare the experimental, also called reference, structure to protein structure models include the root-mean-square deviation (RMSD) [1], the template modeling score [2], the Global Distance Test Total Score (GDT-TS) [3], the Contact Area Difference score (CAD-score) [4], the Local Distance Difference Test (LDDT) [5], and the SphereGrinder score [6]. These and other metrics have been reviewed by Olechnovic et al. [7]. Further, combinations of standard metrics have been developed for the evaluation of the models produced as part of the Critical Assessment of protein Structure Prediction (CASP) experiments [8,9].

In contrast to the metrics which require the experimentally determined structural coordinates of the target sequences, there are a host of quality estimation techniques, such as QMEANDisCo [10], that estimate model quality in the absence of an available reference structure. Such methods are applied in the CASP experiment as well [11], and representative methods are available via the CAMEO-QE website [12]. But we do not discuss their specific applications further here.

To measure the quality of a protein structure model relative to a reference structure, the types of measurements in gross terms include the distance difference between backbone atoms (i.e., *α*-carbons), the distance difference between the sidechain atoms, and the fraction of native contacts. We have previously implemented a complimentary approach called ResiRole [13], which considers comparisons of the functional site predictions made in the models against those made at the corresponding sites in the reference structures. To obtain the functional site predictions, the FEATURE program was implemented [14], which uses information regarding the hydrogen bonding patterns, ionic bonding interactions, hydrophobicity, and other variables to determine the probability of the presence of a functional site, such as a calcium binding site, at given location within a protein’s 3D structure. If the differences between the probabilities of the functional site predictions made in the model versus those made at the corresponding sites in the reference structure are relatively small, then the structure model is deemed to be of high quality and therefore accurate.

Here, we conduct a study comparing the accuracies of protein structure prediction techniques utilized in the Critical Assessment of Structure Prediction experiment, round 15 (CASP15). As with previous CASP experiments, the experimentally determined structural coordinates are not made public until after the structure prediction submissions have been received, and such is done to ensure the fidelity of the competition [15]. One of the key differences between CASP15 and previous iterations of CASP was the high use of AlphaFold-inspired methods [16]; and most groups used AlphaFold models in some way. We report here on the estimated accuracies of the structure prediction techniques based on their abilities to construct models that have the local structural features of the reference structures.

## 2. Materials and Methods 

### 2.1. Calculation of Quality Metrics: The Average and Cumulative PFSSs

Many of the scripts and programs used in this study were written in Python [17] running on the Rocky Linux operating system. These scripts had previously been implemented to run on CAMEO data (See [13]); and modifications were made for the purposes of accounting for input from the CASP data center [18], as implemented for CASP15.

Data regarding each protein model and its corresponding experimental structure first needed to be sorted within the local file system’s directories. The FEATURE program was then run on all the 3D coordinate data files, which were in PDB format. The FEATURE program works by identifying an atom of interest for a particular functional site prediction model and outputs a raw score based on how likely the spherical microenvironment around the atom of interest describes a particular SeqFEATURE functional site [14]. SeqFEATURE functional sites have been derived based on the 3D structural features around atoms of interest that are part of 3D structural motifs. The structure motifs map to primary sequence-based functional site motifs described in PROSITE [19].

Previously, we calculated the mean and standard deviation of the raw FEATURE scores for each SeqFEATURE functional site prediction model using an expansive set of results from approximately four years of CAMEO reference structure data [13]. Assuming that the raw scores are normally distributed, we used the means and standard deviations associated with each of the SeqFEATURE models to calculate Z-scores from the raw scores. The procedures were applied to the functional site predictions made for the CASP15 reference structures and for each of the associated protein structure models.

Data on each of the functional site predictions made for the reference structures with a Z-score greater than the Z-score that corresponded to a specificity level of at least 90% for the corresponding SeqFEATURE model was deemed to meet the threshold for inclusion in the study. That threshold was based on the receiver operating characteristic curve analysis by Toth et al. [13]. The data regarding the Z-scores for the functional site predictions at the corresponding sites in the protein models were also kept. With each of these Z-scores, we calculated the cumulative probability for each functional site prediction [13] using the cumulative density function in SciPy 1.1.0 [20]. The difference score was defined as the absolute value of the difference between the cumulative probability obtained for a functional site prediction in the reference structure and that obtained for the functional site prediction at the corresponding site in the structure model [13]. A similarity score was calculated as one minus the absolute value of the difference score, that is 1 − |Prob(target) − Prob(model)|. The normalized Predicted Functional site Similarity Score (PFSS) for a given functional site prediction was the similarity score divided by the normalization factor gamma.

The normalization factor gamma was the average of the cumulative probabilities of the functional site predictions made for the reference structures. To calculate gamma for a given SeqFEATURE model, all the functional site predictions in the reference structures that met the criteria for inclusion in the study were utilized. The normalization factor, gamma, is shown in the following equation for the calculation of the *PFSS* for a given instance *i* of a functional site prediction.
(1)$${PFSS}_{SeqFeatureMod}^{i}=\frac{{SS}^{i}}{\gamma }$$

The normalization factor gamma helped to address differences in the averages of the similarity scores observed when using each of the different SeqFEATURE models. A gamma was calculated separately for each SeqFEATURE model for each structural domain. Dividing similarity scores by the gamma associated with each SeqFEATURE model for each domain resulted in more normalized PFSS scores across the different types of SeqFEATURE models.

An average PFSS for a given structure modeling group, called group PFSS or gPFSS, was the average across the averages obtained using each of the different SeqFEATURE models. Note the entire process of calculating the gPFSS was done separately for each of the domains modeled by each of the structure prediction techniques. The final group PFSS (gPFSS) for a structure prediction technique was therefore calculated as the average of the gPFSS values obtained across the domains addressed by that modeling technique. An additional metric to assess each modeling group’s performance was the cumulative PFSS (cumPFSS) which was calculated as the sum the gPFSS values calculated for each of the domains modeled by the modeling group.

### 2.2. Description of the CASP15 Dataset Used in the Study

Upon request on 19 January 2023, we obtained permission from the CASP15 organizers to access a dataset regarding the regular targets and the associated protein structure models. We obtained both the experimental structures and structure models for 71 different target protein sequences addressed therein. The targets included 98 domains delineated by their amino acid residue ranges. A full list of the target identifiers, domain identifiers, and domain residue ranges are listed in Appendix A.

In terms of the CASP15 difficulty categories for the 98 domains analyzed in our dataset, 39 were in the free modeling (FM) category, 36 were in the template-based modeling easy (TBM-easy) category, 12 were in the template-based modeling hard (TBM-hard) category, 8 were in the TBM/FM overlap category, and 3 were in the Other category.

To ensure that we mapped all the residues pertaining to each domain properly, the residue ranges for each domain were extracted from the corresponding summary page at the Protein Structure Prediction Center, https://predictioncenter.org/casp15/domains_summary.cgi (accessed on 12 June 2023). Due to gaps in the experimental structures, some residue ranges were discontinuous; that is, they were represented by two or more sets of residue ranges. We identified the residue numbers within each range and mapped them to each domain as found in the coordinate files of the experimental structures and the protein models. The labels for the difficulty category assigned to each domain were also annotated using information from the Protein Structure Prediction Center.

## 3. Results

### 3.1. Description of the Relationships between gPFSS and Standard Quality Metrics

As described in Materials and Methods, for each structure prediction technique, that is each modeling group, the group Predicted Functional site Similarity Score (gPFSS) was obtained by averaging the average PFSSs calculated using each of the different SeqFEATURE functional site prediction models. The procedure for calculating the gPFSS was applied separately for each of the domains modeled by each structure prediction technique, and an average across each domain result was the final gPFSS associated with the structure prediction technique. To describe the relationship between gPFSS and each of the standard quality metrics utilized in the CASP15 experiment, scatter plots were created in which each point represented a result obtained for one of the structure prediction techniques. Figure 1 describes the relationships between gPFSS and the standard metrics, as observed for models in the free modeling (FM) difficulty category.

Figure 1A is a scatter plot of LDDT versus gPFSS. There were 122 structure prediction groups represented for the FM category which addressed one or more of the 39 domains in that category. The Pearson correlation coefficient is 0.98239, which indicates a very strong predictive relationship between LDDT and gPFSS. The *p*-value associated with the correlation coefficient is 2.7872 × 10^−89^. Note that there was considerable variation in the number of domains that were addressed by each group. As examples, the Yang group addressed all 39 domains of the FM dataset while the noxelis group only addressed one domain.

For each modeling group, we calculated the standardized residual in R [21]. The formula for standardized residual was residual/(StDev) × sqrt(1 − hii)), where hii is the leverage for the observation. For the plot shown in Figure 1A, we see one point that corresponded to a standardized residual greater than the absolute value of 3, which indicates it is an outlier based on the three sigma rule [22]. The point corresponds to EMBER3D, which had a standardized residual of −4.33. The result indicates that the LDDT value is much lower than would be predicted based on its corresponding gPFSS. LDDT thereby underestimates the quality of the models produced by EMBER3D. Also, we note that the wuqi method has a high negative standardized residual of −2.96, which indicates that LDDT may also underestimate the accuracy of the models generated by the wuqi structure prediction technique when using the gPFSS as the benchmark.

The reason for the discrepancy for EMBER3D appears to be due to its structure prediction method directly utilizing constraints imprinted by function [23]. We note that as discussed in a study by Weissenow et al., EMBER3D was found to capture effects of sequence variation on structure modification better than methods such as AlphaFold2, which indicates an advance in the understanding protein function [23]. As most of the methods in CASP15 utilized the AlphaFold2 technique in some form, it makes sense that the predictive relationship between LDDT and gPFSS, as obtained using all the groups, would not precisely predict the accuracy of EMBER3D. For EMBER3D, the constraints placed on the structure predictions made by explicitly considering the 3D relationships required for functional sites may thereby have enabled a deviation from that overall predictive relationship.

The scatter plots and the associated predictive relationships between gPFSS and the standard metrics obtained from the CASP15 organizers are given in the Appendix A. Also available are the results when considering only targets in each of the following difficulty categories: All, TBM-easy, TBM-hard, TBM/FM, and Other.

### 3.2. Ranking Modeling Groups Using the gPFSS and cumPFSS Performance Metrics

To estimate the overall performances of the modeling groups, multiple metrics were utilized. As described in Materials and Methods, we calculated the group PFSS (gPFSS) and the cumulative PFSS (cumPFSS) metrics. The value of cumPFSS was the sum of the gPFSS values calculated for each of the domains modeled by the given structure prediction technique. We obtained from the CASP15 organizers the following metrics: GDT-TS, GDT-HA, GDC-SC, GDC-ALL, RMS-ALL, and RMS-CA. These were provided on a per domain basis; and we averaged values across the domains in our dataset, which was comparable to the way in which gPFSS was calculated.

Given that the CASP experiment considers both the quality of the modeled domains and the number of modeled domains as part of assessment criteria of each group’s performance [24], we first reviewed the results of the ranking the modeling groups from the perspective of the cumPFSS metric. Recall that the value of the cumPFSS metric increases as more domains are modeled by a group and as the values of gPFSS for each of the domains increases. Table 1 shows the top 25 groups sorted by the cumPFSS metric. Other metrics shown in the table are the gPFSS metric for each group, the average GDT-TS, and the number of domains that were modeled. Table 1 includes the results obtained regarding all targets across all difficulty categories.

The results using the cumPFSS metric are comparable with the those obtained with the CASP15 score described by Simpkin et al. [25]. For example, within the top 25 groups identified by ranking with the CASP15 score, 15 of those were also in the top 25 when using cumPFSS metric. Notably within the top 5, according to both the cumPFSS metric and the CASP15 score, we see the following groups: UM-TBM, Yang-server, and Yang. As an example of another specific result, when using the cumPFSS metric, PEZYFoldings was ranked 38 while it ranked first when using the CASP15 score [26]. As the cumPFSS is dependent on the number of domains modeled, and the drop in rankings of PEZYFoldings was due in part to the observation that it modeled 96 of the 98 domains in the dataset. When considering groups that modeled 96 domains or more, PEZYFoldings was found to be ranked fourth, behind Yang-Server, UM-TBM, and Yang respectively. To further review the overlap of the ranking results, an extended version of the results displayed in Appendix A is provided in the Appendix A; and it is also available on the ResiRole-CASP15 website at the URL https://protein.som.geisinger.edu/ResiRole-CASP15/ (accessed on 16 October 2023).

Table 2 shows the results of rankings using the cumPFSS metric when considering only targets in the FM category. As with the results seen for all targets, the results obtained when considering only targets in the FM category shows an overlap with top 25 groups identified using CASP15 score. Of the top 25 groups identified by the CASP15 score, as calculated using information on all the targets, 17 groups were also identified within the top 25 identified by the cumPFSS metric as applied to only the FM targets, which constitutes a 68% overlap.

### 3.3. Statistical Comparisons of the Modeling Group Performances

For quantitative comparisons, we review example results obtained when considering all targets. The ranks of RaptorX and OpenFold using the gPFSS metric are compared to their ranks with GDT-TS. See that sorting and ranking with these two metrics can be done on the ResiRole-CASP15 website, https://protein.som.geisinger.edu/ResiRole-CASP15/tables/Metrics_Table_All.jsp (accessed on 16 October 2023), or using provided Appendix A. We observe that RaptorX had a high jump in the rankings using the gPFSS metric, for which its rank was 32, relative to a rank of 46 when using the GDT-TS score. In contrast, OpenFold went down in the rankings as its rank using GDT-TS was 99, and its rank was 102 using gPFSS. More quantitatively, the gPFSS for RaptorX was 0.897 ± 0.08, and we see that RaptorX addressed all 98 domains of the All targets dataset. For OpenFold, its gPFSS was 0.8689 ± 0.096, and it also addressed all 98 domains of the All targets dataset. Upon comparison of the arrays of per domain values of gPFSS for RaptorX and OpenFold with a Mann-Whitney U test, we see that RaptorX has a significantly higher gPFSS, with an associated *p*-value of 0.011. The GDT-TS metric was not sensitive enough to detect the apparently higher rank of RaptorX over OpenFold based on a Mann-Whitney U test when using the arrays of domain averages for the two groups.

For the single, pairwise comparison with the Mann-Whitney U test between RaptorX and OpenFold when using their gPFSS values for the 98 domains that they both addressed, we presume significance based on the observed *p*-value of 0.011. We also performed pairwise head-to-head comparisons of all the groups which modeled all 98 domains for the All targets dataset. These results are given in Appendix A. As there were 52 groups that modeled all 98 domains, there were 51 pairwise comparisons for each of these groups. A Bonferroni correction was therefore applied when considering all these comparisons, which gives a *p*-value threshold of 0.05/51 or 0.00098. Although the *p*-value for the comparison between RaptorX and OpenFold is not below that threshold, we did follow-up studies to verify significance. A Mann-Whitney U test was subsequently used to compare the average PFSS values for the 605 SeqFEATURE models that were obtained for the analyses of the RaptorX and OpenFold structure models respectively. For this comparison of RaptorX and Open Fold the associated *p*-value was 2.23 × 10^−10^, and the result indicates that RaptorX significantly outperformed OpenFold. The results of the pair-wise head-to-head comparisons conducted for all groups which addressed all 98 domains of the all targets dataset are provided in the Appendix A.

### 3.4. Illustrative Examples of the Utility of the PFSS Metric

Examples of functional site predictions made in the reference structures were compared to those made in the structure models in an effort to illustrate the utility of the PFSS metric. Two examples of functional site predictions are described. These sites were selected because the functional site predictions are made in regions of the reference structure that do not have defined secondary structure, that is, they have a random coil or bend designation as indicated by the DSSP program [27].

Figure 2A showcases the CASP15 target with the identifier, T1169-D4. Here we consider our first illustrated example of a functional site prediction. Visualization was done using the ChimeraX program [28]. The target protein is the mosquito salivary gland surface protein 1 (SGS1); and its structure is available, PDB ID: 8FJP [29]. A protein phosphatase 2C site, as predicted using the PP2C.6.ASP.OD2 SeqFEATURE model, is found to be centered at the anchor residue aspartate 1003. The residue resides within the fourth domain of the target. The cumulative probability for the predicted functional site in the reference structure is 0.9935. The raw difference score is −0.0493 for the site in the first Yang Server model submission, and the associated value of the gamma normalization factor for the PP2C.6.ASP.OD2 SeqFEATURE model is 0.9725. The PFSS metric at the residue is therefore (1 − (abs − 0.0493))/0.9724 or 0.9776. Further, using a local implementation of the lddt program [5,30], an LDDT score of 0.5564 was calculated at residue 1003 for the Yang Server model.

Manual review of the sequence centered around residue 1003 appears to match fairly well with the corresponding PROSITE sequence motif that is described at the URL https://prosite.expasy.org/PDOC00792 (accessed on 16 October 2023). The structural environment in the reference structure and the Yang Server model appear to be nearly the same around residue 1003 except for an apparent difference in the rotamer state in the aspartate. As indicated by structure determined by Xu et al. [29], the region lies within the beta-propeller 2 region of the structure. And the beta propeller structural region can contain surfaces implicated in the binding of the phosphorylated protein substrates of the corresponding phosphatase [31]. These findings demonstrate the strong possibility that the predicted phosphatase site is correct. Further follow-up experimental studies are recommended for validation.

Target T1158T-D2 corresponds to multidrug resistance-associated protein 4 (MRP4) [32]. The phosphofructokinase.7.arg.ne SeqFEATURE model predicts there to be a phosphofructokinase site centered on arginine residue 1196, as illustrated in Figure 2B. The first model submissions from OpenFold and EMBER3D are shown aligned with the reference structure. The cumulative probability of the functional site prediction in the reference structure is 0.9945. The PFSS metrics are 0.9851 and 1.0214 for the OpenFold and EMBER3D structure models, respectively. Relatively low LDDT scores of 0.7670 and 0.6244 were found respectively for the OpenFold and EMBER3D models at arginine 1196. Note that although the alignment of the EMBER3D model appears to follow local structure reference structure precisely in the figure it is otherwise displaced from the reference structure because the alignment was done using the entire EMBER3D model with the entire reference structure.

By manual inspection the predicted phosphofructokinase does not appear to be correct. That is based on looking at the corresponding sequence pattern for the phosphofructokinase signature in PROSITE, https://prosite.expasy.org/PDOC00336 (accessed on 16 October 2023). There does not appear to be a strong sequence match of the sequence pattern and the sequence of in the MRP4 protein. Nonetheless, the functional site predictions in the target structure and the models do appear to aid with assessing the quality of the models.

## 4. Discussion

### 4.1. Comparison of gPFSS to Standard Metrics

There is a need in the field of protein quality estimation to develop methods that evaluate the quality of protein models according to direct comparisons between experimental structures and structure models. Traditional metrics include those that are distance dependent, such as the RMSD, or dependent on preserved contacts. The ResiRole method described here considers the recounting of the constellation of structural features associated with functional site predictions within the structure models relative to the experimental structures as a complementary approach.

ResiRole examines the precise orientation of the structural features at predicted functional sites, as extracted by the FEATURE program, for each of the selected residues that in theory may be part of actual functional sites. The structural features examined by the FEATURE program considers many types of complementary structural features such as the number of hydrophobic atoms, the number of polar atoms, and the number of atoms from each type of secondary structure element, to name a few. The constellation of these structural features as found at functional sites constitute a relatively unique composition around the site that has been selected as a central point in the functional site three-dimensional motif. As this unique combination of structural features is only present at the associated functional site and a probability of the functional site given a set of structural features can be calculated, the measurement of the similarity of the functional site prediction probabilities between those found for the experimental structure and the structure model constitutes a way to measure the degree to which the experimental structure and the model are similar. A more robust measure of similarity is obtained when hundreds of different types of functional site predictions, as made available through the application of hundreds of different SeqFEATURE models, are applied.

As described in the study, we derive the gPFSS metric as an overall similarity metric using the results obtained via the application of the different SeqFEATURE models. The gPFSS metric was found to correlate well with the standard quality assessment metrics, as indicated in Figure 1. We may further consider the reasons for the linear relationships observed when utilizing domains from FM difficulty category. For the FM category, we observed a precise linear relationship between LDDT and gPFSS; and the associated Pearson correlation coefficient was 0.9824. We postulate that when using template structures for modeling they may have functional sites well described; but these functional sites may not be part of actual functional sites in the target structure. When calculating the difference in probabilities for these sites in the models versus the reference structures, they may reach the threshold for inclusion in the study. But these sites would likely introduce error in the measurement of the average PFSS. With a focus on targets from the FM category, which do not have template structures with high sequence similarities to the target sequences, we may eliminate some noise when calculating the PFSS. In doing so, the correlation between the LDDT metric and gPFSS is found to be the strongest. Further, the final group ranking by the cumPFSS metric for the FM targets had a 68% overlap with the ranking of the top 25 groups made with the CASP15 score, which indicates that accuracy is measured well when using the gPFSS metric as applied to the FM targets.

As LDDT is one of the standard metrics for benchmarking the accuracies of structure prediction techniques [12], our finding of a highly predictive linear relationship between LDDT and gPFSS indicates that gPFSS can be used as an effective benchmarking metric as well. The added value of the gPFSS appears to be in identifying cases where expected gPFSS value was not close to its predicted value based on its predictive relationship with LDDT. Recall the result that EMBER3D was a statistical outlier according to that relationship.

An added value of gPFSS is that it can assess how well a structure prediction technique can be used to construct models that have the structural features of predicted functional sites found in experimental structures. We extrapolate that the gPFSS can be used as a proxy for assessing the ability of a structure prediction method to generate models that have the structural features required for actual functional sites. The application of the gPFSS metric may, in turn, be used to guide the development of structure prediction techniques, with the goal of having further optimization that ensures accurate prediction of the structures at predicted functional sites.

For our illustrative examples, which are shown in Figure 2A,B, we consider cases where a standard quality metric, LDDT, assigns a relatively low score. In contrast the PFSS indicates a relatively high-quality structure model at the specified site. These examples were also selected so that the anchor residues of the functional site predictions were centered within coil or loop regions. Such was done to further illustrate the added value of the PFSS metric since loop regions are challenging in terms of accurate structure prediction [31,32]. Having multiple sensitive measures for assessing model quality at especially difficult regions to model, such as loops, may thereby aid the development of the structure prediction methods.

### 4.2. Further Descriptions of Individual Group Performances

Consider the result that RaptorX was found to outperform OpenFold according to gPFSS and cumPFSS metrics. Functional constraints are directly used to guide the structure predictions with RaptorX. RaptorX uses CATH domains to build both the template-based and template-free models of domains [33]. RaptorX is based in part on information from CATH S35 sequence clusters. The sequences represented in the CATH S35 clusters have less than 35% sequence identity with each other; but the clusters are curated to contain functionally similar sequences. The associated functional similarity criteria are met using FunFams, which are homologous superfamilies likely to have highly similar structures and functions [34]. Essentially the sequences used to build the models are clustered into families, and sequences with dissimilar functions are removed or filtered out. 3D structure modeling then proceeds using the corresponding constraints imposed by function. In contrast, OpenFold, which is a fast and trainable implementation of AlphaFold2 [35], uses evolutionary related sequences to build the structure models; but it does not specifically filter out sequences that are functionally unrelated. We infer that direct use of the constraints imposed by function, as utilized by RaptorX but not by OpenFold, appears to have given RaptorX a boost in its ranking and performance as assessed with the gPFSS metric.

As a general trend for modeling groups that explicitly utilize functional constraints to build their models, their ranks using the gPFSS metric were higher than their ranks with GDT-TS. Consider as examples the modeling groups Cerebra, Gonglab-THU, EMBER3D, RaptorX, and BhageerathH-Pro. Each had an increased rank with gPFSS. According to method descriptions in the CASP15 Abstract book, https://predictioncenter.org/casp15/doc/CASP15_Abstracts.pdf (accessed on 16 October 2023), the Cerebra method developed by Haipeng Gong’s laboratory uses a subset of CATH S35 v4.21 [36]. There are subsequent searches for homologous sequences to build multiple sequence alignments as input for the generation of the structure prediction models. Cerebra ranked 122 with gPFSS and 123 with GDT-TS. Another method from the Dr. Gong’s laboratory, Gonglab-THU, ranked 121 with gPFSS while its rank with GDT-TS was 126. As mentioned above, regarding RaptorX, the use of constraints imprinted by function via the incorporation of CATH S35 sequence families may assist in boosting the rank of the modeling group when its ranking is done using gPFSS is compared to its rank obtained with GDT-TS. That is because the gPFSS metric is focused on assessing the accuracy of recounting the structural features at functional sites.

EMBER3D’s rank was 112 with gPFSS while its rank with GDT-TS was 119. EMBER3D directly uses structural constraints imposed by function in its modeling methodology [Weissenow, 2022 #860], which we infer may account for its increase in rank. The BhageerathH-Pro group rose from a rank of 126 with the GDT-TS to a rank of 120 using gPFSS. According to method descriptions in the CASP15 Abstract book, the BhageerathH-Pro method developed by Dr. Jayaram and coworkers uses as part of its methodology Mask BLAST [Pathak, 2021 #882]. Mask BLAST is utilized to aid with the functional assignment of the input sequence. The use of that step for structure prediction may help account for the gain in the performance rankings when using the gPFSS metric relative to when using GDT-TS.

### 4.3. Applications for Guiding Functional Characterizations

The first illustrated example discusses a possible protein phosphatase 2C site in the mosquito salivary gland surface protein 1. This example describes a structure model that exhibited enough of the structural features at the predicted functional site to enable a very high probability prediction. Based on our manual review of the likelihood of the site, we further find evidence that it is a correct prediction. We recommend that follow-up studies be conducted to possibly identify any of the phosphoprotein substrates and fully verify the functional relevance of the predicted functional site. Such may be done by possibly mutating positions in the site and noting any observed changes in the protein’s function.

In general, the identification of a putative and likely functional sites lends credence to the conclusion that the Yang Server technique, and likely the other techniques represented in CASP15, produce structure models that are accurate enough for robust functional site prediction. Noting this important inference, we are working to provide as an additional resource the full set of functional site predictions. Their probabilities in the reference structures will be given along with their difference scores for each of the structure models.

### 4.4. Limitations of the Described Methodology

One limitation of the ResiRole method is that quality assessment is only based on the findings at functional site predictions at specific residues within the structure models. Other metrics such as LDDT and GDT-TS have quality measurements available at all residues within the structure models. Performance of a modeling technique is therefore best assessed with all available functional site predictions across all the models of a given difficulty category as that provides a more accurate mean performance measure. Further, as described in reference to Figure 2B, the functional site prediction, even if it has a high cumulative probability, may not actually correspond to a functional site that can be validated experimentally. So further manual review of the likelihood of the functional site prediction is recommended prior to conducting experimental studies. We although surmise that the functional site predictions made with high associated probabilities can indeed serve as a measure of structure model quality in the absence of the functional site being present. That is because the similarity of the functional site predictions in the experimental structure and structure model reflects the degree to which the structural features of the experimental structure are represented in the structure model.

### 4.5. Possible Future Directions

The current basis of the ResiRole method is the use of functional site predictions as made with the FEATURE program. In theory, other ways to predict functional sites could be used in a similar manner if their scores can be converted to accurate probabilities. As described by Toth et al. [13], the conversion of the scores generated by the FEATURE program into probabilities was based on the extensive benchmarking studies where Z-scores could be generated for each type of function. These results were kindly provided via a personal communication by Mike Wong of the Helix group at Dr. Russ Altman’s laboratory. The Z-scores were converted to cumulative probability estimates through the cumulative density function in SciPy as described in Toth et al. In theory, that entire process can be repeated with different types of new functional sites and, as such, the application of other methods to predict function based on structure constitute a future research direction. The goal would be to more comprehensively address the various types of functions that can be predicted from three-dimensional coordinates and thus be used to assess the quality of protein structure models.

Further, consider that the CASP15 score described by Simpkin et al. [Simpkin, 2023 #869] is based on Z-scores that range outside of 0–1, so some domain evaluation units may have a relatively large effect over others as part of the CASP15 cumulative metric. The cumPFSS metric, on the other hand, is normalized to be between zero and approximately one, so that any one domain can only contribute to the cumPFSS metric by no more than approximately a value of one. That may minimize the uneven influence of the scores that some domains may have over others. We infer that the normalization procedure undertaken to develop the cumPFSS improved its robustness in accurately ranking the modeling groups. In theory, the CASP15 score can be converted to a cumulative probability estimate of performance as it is itself a combination of Z-scores.

## 5. Conclusions

Normalized benchmarking metrics for the evaluation of protein structure models are described that include the gPFSS and cumPFSS metrics. The gPFSS metric correlates exceptionally well with standard assessment metrics. A high gPFSS indicates that the structure prediction technique would likely produce structure models that have a high agreement with the functional site predictions made in the experimental structures. As such, the gPFSS metric is proposed as a complementary means to assess the quality of individual structure models.

When used to compare structure prediction techniques, the gPFSS metric and cumPFSS appear able to identify which techniques produce structure models better at recounting the constellation of structural features required for function that are seen in the experimental structures. The gPFSS can help identify which modeling techniques have had their accuracies either over or underestimated relative to the standard assessment metrics. Further, the gPFSS metric provides a sensitive metric that can be used to compare structure prediction techniques in a head-to-head manner to see which technique performs better in quantitative terms.

One goal of providing the results of the study in the form of Appendix A, such as the per domain results, is to further support the development of the structure prediction techniques as the gPFSS may be used as an additional benchmarking metric. With the further optimization of the structure prediction techniques, we anticipate that the availability of highly accurate structure models may further enable applications such as structure-based drug design [37].

When looking at specific functional site predictions, an added value of using the PFSS metric is to identify local areas of a model that are deemed to be highly accurate. That information can be paired with knowledge of the functional site prediction probabilities as they are found in the corresponding reference structures. The combined information may serve as a guide for further functional characterization of the protein. Such can be done with additional modeling studies and follow-up experimental studies that validate the predicted functions. As an additional goal, we are working to provide all of the functional site predictions that are of high probability in the reference structures along with models that are assessed to be of high accuracy.

## Figures and Tables

**Figure 1 bioengineering-10-01377-f001:**
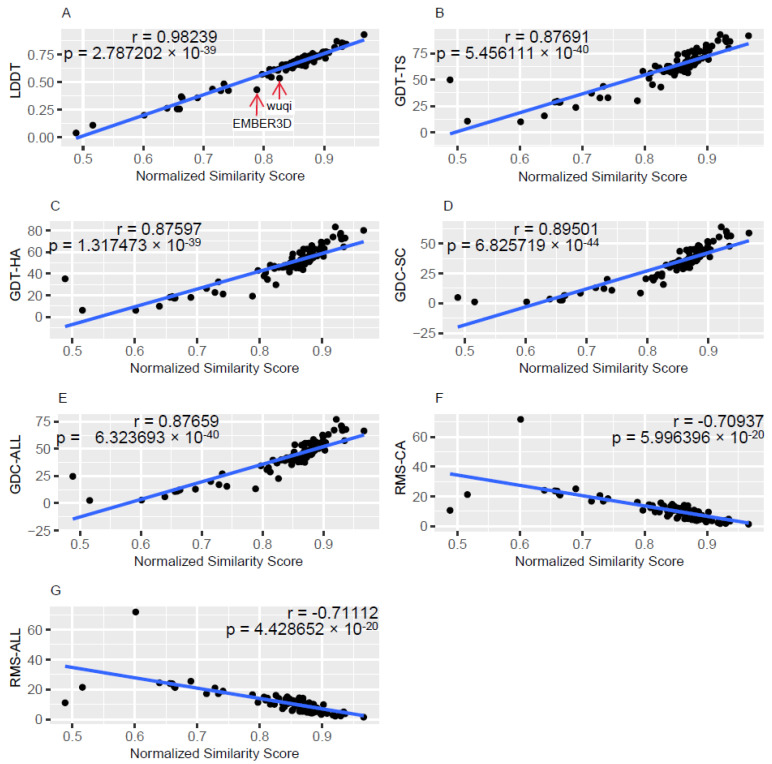
Scatter plots of the standard quality metrics used in CASP15 versus the corresponding group Predicted Functional site Similarity Score (gPFSS). The results are for domains of the FM category. (**A**) Local Distance Difference Test, LDDT. (**B**) global distance test total score, GDT-TS. (**C**) global dis-tance test high accuracy, GDT-HA. (**D**) global distance calculation for side chains, GDC-SC. (**E**) glob-al distance calculation all atoms, GDC-ALL. (**F**) alpha carbon root-mean-squared deviation, RMS-CA. (**G**) all atom root-mean-squared deviation, RMS-ALL.

**Figure 2 bioengineering-10-01377-f002:**
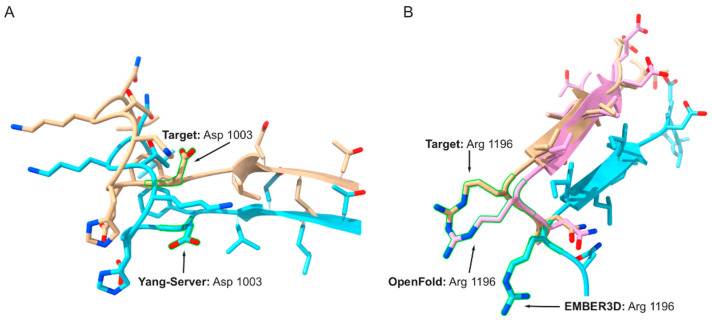
Illustrative examples of functional site predictions. (**A**) Target, T1169T-D4, can be seen above on the left. The target’s reference structure (tan) is aligned with a first structure model produced by the Yang-Server (cyan). As seen in both structures, the aspartic acid residue that serves as the anchor residue of the functional site motif (Asp 1003) is outlined in green. (**B**) Alignments of two models representing target T1158T-D2 with the reference structure are shown. The residue outlined in green is the anchor residue, arginine 1196 (Arg 1196). The target’s reference structure (tan) is slightly covered by the first attempt submission from OpenFold (magenta). The first attempt submission for EMBER3D (cyan) is also shown, and it also has been aligned to the reference structure.

**Table 1 bioengineering-10-01377-t001:** Ranked list of modeling groups according to cumulative PFSS when considering the domain targets from all difficulty categories. The group PFSS, the average GDT-TS, and the total number of domains modeled by each modeling group are also indicated.

Group Name	cumPFSS	gPFSS	Average GDT-TS	Domains
UM-TBM	89.1575	0.9098 ± 0.069	83.8797 ± 16.125	98
Yang_Server	89.0518	0.9087 ± 0.069	84.2358 ± 15.183	98
Yang	88.842	0.9066 ± 0.068	83.3322 ± 17.207	98
MULTICOM_refine	88.114	0.8991 ± 0.086	79.012 ± 21.53	98
MULTICOM_qa	88.0744	0.8987 ± 0.086	78.4293 ± 21.723	98
Kiharalab	88.0307	0.8983 ± 0.086	79.5234 ± 19.03	98
MULTICOM_deep	87.9948	0.8979 ± 0.087	78.6633 ± 21.771	98
MULTICOM	87.9889	0.8978 ± 0.082	80.1332 ± 20.214	98
RaptorX	87.9075	0.897 ± 0.08	77.5924 ± 21.971	98
MULTICOM_egnn	87.871	0.8966 ± 0.087	78.6379 ± 21.508	98
server_126	87.851	0.8964 ± 0.08	76.6975 ± 22.13	98
BAKER	87.8253	0.8962 ± 0.072	78.9276 ± 19.247	98
GuijunLab_DeepDA	87.5177	0.893 ± 0.09	76.0575 ± 22.965	98
McGuffin	87.5162	0.893 ± 0.089	79.6776 ± 21.1	98
Bhattacharya	87.4844	0.8927 ± 0.089	72.7336 ± 23.766	98
GuijunLab_Threader	87.4766	0.8926 ± 0.092	75.4201 ± 23.506	98
MUFold	87.4764	0.8926 ± 0.088	75.7452 ± 23.314	98
server_124	87.4566	0.8924 ± 0.081	76.6425 ± 22.425	98
Manifold	87.4383	0.8922 ± 0.092	78.3286 ± 21.427	98
Agemo_mix	87.4285	0.8921 ± 0.092	75.4971 ± 24.464	98
Manifold_E	87.3873	0.8917 ± 0.092	78.4974 ± 21.096	98
DFolding	87.2864	0.8907 ± 0.088	79.921 ± 21.185	98
DFolding_server	87.2661	0.8905 ± 0.089	77.5 ± 23.235	98
AP_1	87.2457	0.8903 ± 0.078	72.0505 ± 25.516	98
bench	87.2254	0.8992 ± 0.078	79.1023 ± 19.739	97

**Table 2 bioengineering-10-01377-t002:** Ranked list of modeling groups according to cumulative PFSS when considering the domain targets from the FM category. The group PFSS, the average GDT-TS, and the total number of domains modeled by each modeling group are shown.

Group Name	cumPFSS	gPFSS	Average GDT-TS	Domains
Yang	34.9172	0.8953 ± 0.079	76.1839 ± 21.087	39
UM-TBM	34.885	0.8945 ± 0.078	77.771 ± 19.278	39
bench	34.3215	0.88 ± 0.094	70.311 ± 24.287	39
Kiharalab	34.2139	0.8773 ± 0.106	70.85 ± 23.167	39
MULTICOM	34.0653	0.8735 ± 0.102	69.8126 ± 24.293	39
MULTICOM_refine	34.0649	0.8735 ± 0.107	67.9574 ± 26.441	39
BAKER	34.0433	0.8729 ± 0.092	69.1556 ± 21.431	39
Yang_Server	33.9965	0.8946 ± 0.083	79.6832 ± 17.871	38
MULTICOM_qa	33.9227	0.8698 ± 0.107	67.4618 ± 26.177	39
server_126	33.9147	0.8696 ± 0.097	63.9297 ± 26.457	39
MULTICOM_egnn	33.9086	0.8695 ± 0.109	68.0495 ± 26.275	39
Shennong	33.8923	0.869 ± 0.104	65.7933 ± 25.358	39
Wallner	33.8653	0.8683 ± 0.106	66.4941 ± 28.466	39
McGuffin	33.7385	0.8651 ± 0.103	69.0397 ± 23.96	39
DFolding	33.7349	0.865 ± 0.109	70.3521 ± 24.097	39
RaptorX	33.7334	0.865 ± 0.098	65.0223 ± 26.525	39
server_124	33.6896	0.8638 ± 0.103	63.47 ± 27.09	39
MUFold_H	33.6521	0.8629 ± 0.103	63.9087 ± 27.151	39
Bhattacharya	33.621	0.8621 ± 0.108	59.8569 ± 27.209	39
AP_1	33.5958	0.8614 ± 0.091	56.4226 ± 28.276	39
Asclepius	33.5758	0.8609 ± 0.11	63.1954 ± 28.892	39
Manifold	33.5282	0.8597 ± 0.117	66.6654 ± 26.378	39
DFolding_server	33.5281	0.8597 ± 0.108	65.7005 ± 28.139	39
PEZYFoldings	33.5064	0.8817 ± 0.095	74.9637 ± 22.453	38
hFold_human	33.4941	0.8588 ± 0.112	64.0833 ± 27.371	39

## Data Availability

Additional access to the tabular results are available vit the following URL: https://protein.som.geisinger.edu/ResiRole-CASP15 (accessed on 16 October 2023).

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
