# Peer review of "Assessment of the Performances of the Protein Modeling Techniques Participating in CASP15 Using a Structure-Based Functional Site Prediction Approach: ResiRole"

_bioengineering, 2023, doi:10.3390/bioengineering10121377_

Round 1

Reviewer 1 Report

Comments and Suggestions for Authors

This study focuses on the importance of assessing the quality of protein structure prediction models. It introduces the "ResiRole" method, which evaluates model quality by examining how well the models preserve the structural characteristics associated with functional sites. The authors applied ResiRole to benchmark the performance of modeling groups in the Critical Assessment of Structure Prediction (CASP) experiment, round 15. They calculated a metric called the Predicted Functional site Similarity Score (PFSS) to measure model quality, which correlates strongly with traditional quality metrics. The results indicate that the PFSS metric is effective in ranking modeling groups based on their ability to accurately represent structural features at predicted functional sites. The study also highlights the impact of functional constraints on certain modeling methods, such as EMBER3D. The methods are presented in sufficient detail to allow for its replicability. Therefore, I recommend to accept the manuscript after the following modifications:

1.   A comparison with the available prediction methods should be elaborated.

2.   How much your method effective in loop modelling and side chain conformation assignment?

3.   How robust is the correlation between the gPFSS metric and standard quality metrics like LDDT and GDT-TS?

4.   Provide a detailed explanation of the steps involved in calculating the Predicted Functional site Similarity Score (PFSS).

5.   Some experimental support is missing to validate the hypothesis. Authors should complement strongly with the available literatures.

6.   What is the overall impact of this research on the field of protein structure prediction and related applications?

Comments on the Quality of English Language

NA

Author Response

Dear Reviewer #1:

First and foremost, we would like to thank you for your time and efforts in reviewing our manuscript. We are grateful for your constructive comments that have led to a better result.

Enclosed please find our revised manuscript titled "Assessment of the Performances of the Protein Modeling Techniques Participating in CASP15 Using a Structure Based Functional Site Approach: ResiRole" by Geoffrey Huang, Thomas K. Parry, and William A McLaughlin.

We have addressed all your comments to the best of our ability and revised the manuscript accordingly. Please see our summarized responses to your comments below.

Reviewer 1 Minor Remarks:

  1. A comparison with the available prediction methods should be elaborated.

We are unsure of how to interpret this recommendation. If the intent is for us to describe the performance of our structure prediction method, please know that we do not have one. The CASP experiment provides sequences to groups who have developed structure prediction techniques, each group develops three-dimensional structure models based on the primary sequences. They send these to the CASP organizers. The organizers then compare these models to experimental structures which that have held confidentially. At the CASP conference the results of the quality assessments and, where allowable, three-dimensional coordinates of both the structure models and the experimental structures are displayed. Our method complements the methods that are used to assess the quality of the structure models through the comparisons with the experimental structures.

Another alternative interpretation of the recommendation is to possibly compare PFSS to the predicted LDDT scores (pLDDTs). We do plan to do that in the future. Our focus here is to describe a new quality assessment metric, the PFSS. The metric is compared to standard quality metrics which measure the accuracy of the structure models by directly comparing them to the experimental structures. As the PFSS falls into the category of metrics which directly use and compare the structure models to the experimental structures, the standard quality metrics in this category, such as LDDT and GDT-TS, are considered. The goal is to demonstrate the utility of the PFSS metric to complement the standard metrics for the evaluations of the quality of the structure models after the fact, that is after the experimental structures have become available.

  1. How much your method effective in loop modelling and side chain conformation assignment?

Thank you for the questions. Again, we do not have a structure prediction technique.

To illustrate the application of the method to evaluate loop quality at identified functional site predictions, examples were located that were within loops. Loops were identified by reviewing the output of the mkdssp program as applied to the reference structures. In the output files each residue was assigned to a specific secondary structure. Loops were identified as regions of bends or no secondary structure which indicated random coil.

We reviewed the raw difference score at this location in relation to the LDDT score that was generated for the respective residue. We were able to find instances where we see a lower LDDT value (0.5-0.6) and a high PFSS. Two illustrative examples are selected which describe the identification of highly accurate structures based on the PFSS. As the locations were selected to have low LDDT scores, they illustrate the complementarity of the approach which in turn may further aid the development of the structure prediction techniques by providing an additional benchmark to optimize on.  

  1. How robust is the correlation between the gPFSS metric and standard quality metrics like LDDT and GDT-TS?

A strong correlation between the gPFSS vs the standard quality metrics is described. See Figure 1A where the correlation coefficient was 0.98239 for the plot of PFSS versus LDDT. To get the quality metrics for the comparisons to PFSS, we used official CASP documents obtained from the website to calculate the standard quality metrics that pertained to our study. We made sure to focus only on the domains that the ResiRole function had results, and we calculated the standard metric averages per participating group. It is worth noting that the inconsistency of domains being analyzed could play a factor in the strength of the correlation. Some groups modeled 98 domains while others modeled less. 

  1. Provide a detailed explanation of the steps involved in calculating the Predicted Functional site Similarity Score (PFSS).

In order to calculate the PFSS score we need to perform the following steps: 1. Obtain a difference score. The difference score was then defined as the absolute value of the difference between the cumulative probability obtained for a functional site prediction in the reference structure versus that obtained for the functional site prediction at the corresponding site in the structure model. 2. Convert this to a similarity score. A similarity score was then calculated as one minus the absolute value of the difference score. 3. Calculate gamma, which was used as a normalization factor. Gamma was the average of the cumulative probabilities of the functional site predictions made for the reference structures per difficulty. To calculate gamma for a given SeqFEATURE model, all the functional site predictions in the reference structures which met the criteria for inclusion in the study, that is had a Z-score that was higher than the Z-score which corresponded to the 90% specificity level, were utilized. And 4. Calculate the PFSS. The normalized Predicted Functional site Similarity Score (PFSS) for a given functional site prediction was the similarity score divided by the normalization factor gamma.

  1. Some experimental support is missing to validate the hypothesis. Authors should complement strongly with the available literatures.

The CASP experiment is conducted in a manner such that the experimental structures are made available to the structure prediction groups after the structure predictions have been submitted. The ResiRole method applied here to the CASP15 results assesses how well the structure prediction groups did in relation to the experimentally derived structures. Some of these structures are available in the PDB and have corresponding journal articles. We reference these structures in the text describing Figure 2A and Figure 2B. The remaining experimental structures are made available to the CASP organizers and to the participants for assessment purposes. 

  1. What is the overall impact of this research on the field of protein structure prediction and related applications?

As each CASP experiment builds upon previous results, we aim for the structure prediction community to use the results provided to further aid method development. Knowledge of which models are predicted with highest accuracy by the PFSS can help guide the development of methods that are able to better predict protein structures at and near functional site predictions. We surmise the PFSS is sensitive to small changes in structure which lead to relatively large changes in functional site probabilities. We encourage the structure prediction groups to use the PFSS as a benchmarking metric to aid the development of structure prediction methods.

Sincerely yours,

William A. McLaughlin, Ph.D.

Reviewer 2 Report

Comments and Suggestions for Authors

The authors describe the performance of a new programme, ResiRole, that evaluates the quality of structure predictions of proteins in comparison with established methods. 

In contrast to standard methods ResiRole focuses on the evaluation of how well functional sites within the protein are modelled in comparison to the experimental structure. The functional sites themselves need to be predicted and characterised for this approach. For this purpose the authors use the FEATURE programme. FEATURE predicts functional sides and associates a lot of characteristics to it like the position of hydrophobic and hydrophilic groups and others. 

The comparison between the quality assessment of ResiRole and the standard tools shows that there is a very high correspondence between them (see figure 1) when applied to proteins in the references set from CASP15 - a contest hold to evaluate structure prediction models. 

However, there are individual outliers. 

The differences might be explained by the underlying training methods used. Standard methods build structure families by evolutionary criteria - ResiRole by functional similarity. When comparing the scores of how well a model corresponds to its experimental structure ResiRole scores are more similar to other tools focussing on functional sites than tools using evolutionary related sequence groups. 

The authors conclude that because of the overall similar results ResiRole can be used to evaluate the quality of a programme to correctly predict the three dimensional structure of a protein. The programme focuses on functional sites. A good ResiRole score indicates that the evaluated programme is able to model functional sites well in agreement with programmes that predict functional sites within protein sequences. 

I think the article can be published as it is. The authors describe the principles of how ResiRole calculates its score. The programme adds another flavour to the evaluation of structure prediction programmes - a focus on functional site modelling. The comparison with standard techniques are well presented and the differences in the scores well explained. 

Author Response

Dear Reviewer #2:

We would like to thank you for your time and effort in reviewing our manuscript. Please see that we considered further your interpretation of the conclusions in the fourth sentence of the Conclusion section.

Enclosed please find our revised manuscript titled "Assessment of the Performances of the Protein Modeling Techniques Participating in CASP15 Using a Structure Based Functional Site Approach: ResiRole" by Geoffrey Huang, Thomas K. Parry, and William A. McLaughlin.

Sincerely yours,

William A. McLaughlin, Ph.D.

Reviewer 3 Report

Comments and Suggestions for Authors

Computer modeling of protein structures plays a huge role in the modern world. Which naturally necessitates the development of accompanying algorithms for checking the quality of models.

It seems to me personally that classical methods for assessing the quality of a protein structure model relative to a reference structure would be sufficient. However, new approaches, including the one proposed by the authors, are constantly emerging. In the proposed approach, “functional sites” are selected according to certain criteria and the similarity of the model and structure across these sites is compared. Because selection takes into account potential intramolecular contacts (the hydrogen bonding patterns, ionic bonding interactions, hydrophobicity), then in theory these will be regions whose modeling will give higher accuracy than the average for the protein. If this is not the case, then something is wrong with the selection. Because it is well known that modeling well-structured areas is practically no longer a problem, while the accuracy of modeling drops dramatically in the absence of a good spatial structure. Therefore, the development of evaluation algorithms in the proposed direction seems to me rather limited. Since it comes down to determining the areas of the most accurate modeling and assessing the accuracy of modeling such areas. The results will, on average, be consistent with those obtained in other assessment systems.

Moreover, when reading the article there is a feeling of some kind of impersonality of the text, as if it was written by a computer. The situation could be improved if there was at least some visibility. Is it possible to give an example of at least one modeling problem with Figures showing bad and good modeling based on the results of different evaluation algorithms?

Minor

Abstract

I think something wrong with the sentence in lines 18-20

Charter 2.2

I didn't find any tables in the supplement files. And it's hard for me to figure out how to relate 71 proteins and 98 domains.

Сonclusion

The Сonclusion consists of numerous repetitions of two facts in different words: lines 423-426, lines 426-429, and lines 430-433. Some phrases do not make any sense, for example: lines 420-422.

Comments on the Quality of English Language

Some phrases are difficult to understand

Author Response

Dear Reviewer #3:

First and foremost, we would like to thank you for your time and efforts in reviewing our manuscript. We are grateful for your constructive comments that have led to a better result.

Enclosed please find our revised manuscript titled "Assessment of the Performances of the Protein Modeling Techniques Participating in CASP15 Using a Structure Based Functional Site Approach: ResiRole" by Geoffrey Huang, Thomas K. Parry, and William A. McLaughlin.

We have addressed all your comments to the best of our ability and revised the manuscript accordingly. Please see our summary and summarized responses to your minor comments below.

Major remarks

Regarding your initial comments about the application of the technique to evaluate the quality of the models at regions that are difficult to model, please see the new section in the results, section 3.4. Here we consider the evaluation of regions in the reference structure that are designated as loops or random coils, which are known to be regions that are difficult to accurately model. We find that the PFSS at predicted functional sites in these regions appear to complement the measurements with the standard quality metric LDDT. For these analyses, we ran a local installation of the lddt program. We compared the resulting LDDT scores with the PFSS values and identified instances where the LDDT score was low, yet the PFSS was high, which illustrates the complementarity of the metrics. 

Minor remarks

  1. I think something wrong with the sentence in lines 18-20

Thank you for identifying the issue. We rewrote the sentence to be the following.

To gauge model quality, a normalized Predicted Functional site Similarity Score (PFSS) was calculated as the average of one minus the absolute values of the differences in the functional site prediction probabilities, as found in the experimental structures versus those found at the corresponding sites in the structure models.

  1. I didn't find any tables in the supplement files. And it's hard for me to figure out how to relate 71 proteins and 98 domains.

We apologize for the error in our submission regarding section 2.2. We have reuploaded the tables in the supplementary files for your reference. When these materials become available to you, the descriptions of how the CASP organizers split the protein targets by various domains will become clear, as seen in supplemental table #1.

  1. The Сonclusion consists of numerous repetitions of two facts in different words: lines 423-426, lines 426-429, and lines 430-433. Some phrases do not make any sense, for example: lines 420-422.

Please see that we attempted to rewrite the last few sentences of the Conclusion section to decrease repetition of the same concepts.

  1. Some phrases are difficult to understand.

Please see that we did copy editing to improve ease of interpretation. 

Sincerely yours,

William A. McLaughlin, Ph.D.

Round 2

Reviewer 1 Report

Comments and Suggestions for Authors

I appreciate the effort you put into revising your manuscript, but still I am not convinced that it meets the high standards of the Journal. I recommend that you consult with a more experienced researcher in your field to get feedback on your manuscript.

1. The revised manuscript should be well-organized and easy to read. The font should be clear and readable in single color, and the margins should be generous. The headings and subheadings should be clear and concise, and the text should be free of errors in grammar and spelling.

2. The authors should strive to improve the language of the revised manuscript. This includes using clear and concise language, avoiding jargon, and ensuring that the manuscript is free of errors in grammar and spelling.

3. You also make sure that the figures are clear, well-labeled, and relevant to the text. The figures should also be of high quality, with good resolution and contrast.

4. Authors shoudl discus the implications of the findings in more detail. The authors should also address any limitations of the study and suggest future research directions.

5. Figure 2.  should be imporved. No black background. Proper labelling and presentation required.

Comments on the Quality of English Language

I found minor issue, many typos and grammatical errors are seen in the paper. There are grammatical mistakes and typographical errors in the manuscript. The author should recheck this manuscript carefully and remove all such errors.

Author Response

November 27, 2023

Dear Reviewer #1:

We would like to thank you again for your constructive comments and taking the time to further review our manuscript.

Enclosed please find our revised, round two, manuscript titled "Assessment of the Performances of the Protein Modeling Techniques Participating in CASP15 Using a Structure Based Functional Site Approach: ResiRole" by Geoffrey Huang, Thomas K. Parry, and William A. McLaughlin.

We have addressed all your comments to the best of our ability and revised the manuscript accordingly. Please see our summary and summarized responses to your comments below.

Major remarks

  1. I appreciate the effort you put into revising your manuscript, but still I am not convinced that it meets the high standards of the Journal. I recommend that you consult with a more experienced researcher in your field to get feedback on your manuscript.

We were able to contact an expert in the field to gain further insight as to the importance of the work. In addition to further describing the method to assess modeling group performance, we are now emphasizing the results regarding the functional site predictions. The functional site predictions can enable follow-up studies aimed at further characterizing the proteins experimentally, which we further describe (subsection 4.5). 

  1. The revised manuscript should be well-organized and easy to read. The font should be clear and readable in single color, and the margins should be generous. The headings and subheadings should be clear and concise, and the text should be free of errors in grammar and spelling.

Please know that we went through and tried to improve the flow of the manuscript. We made sure to utilize and cross check the format of the template and other published manuscripts in Bioengineering.

  1. The authors should strive to improve the language of the revised manuscript. This includes using clear and concise language, avoiding jargon, and ensuring that the manuscript is free of errors in grammar and spelling.

Please see that when using acronyms in the abstract we first gave the full name followed by the acronym (Ln 24). That was done to help ensure the reader could better comprehend the jargon used.

  1. You also make sure that the figures are clear, well-labeled, and relevant to the text. The figures should also be of high quality, with good resolution and contrast.

Please see that we provide high quality versions of the figures as part of the zipped files. If there is any figure that requires a better resolution or quality, please let us know otherwise.

  1. Authors should discuss the implications of the findings in more detail. The authors should also address any limitations of the study and suggest future research directions.

We further discuss the implications of the findings in terms of guiding further functional characterizations of the proteins in sections 4.3 (Ln 463). In addition, we have now incorporated a discussion subsection towards the limitations of the ResiRole tool (section 4.4 (Ln 479)). Lastly, we attempt to describe potential future research applications in section 4.5 (Ln 495) to aid the readers in understanding the impact that the research could lead to.

  1. Figure 2.  should be improved. No black background. Proper labelling and presentation required.

We changed the background of Figure 2 to white and incorporated arrows to enable more precise labeling.

  1. I found minor issue, many typos and grammatical errors are seen in the paper. There are grammatical mistakes and typographical errors in the manuscript. The author should recheck this manuscript carefully and remove all such errors.

We went through the manuscript to further resolve errors in grammar and format. Thank you for the feedback.

Sincerely yours,

William A. McLaughlin, Ph.D.

Reviewer 3 Report

Comments and Suggestions for Authors

The authors took the comments seriously and made the required changes and additions to the article, including a figure illustrating examples of predicted structures and explanations for them.

Minor.

My additional tips for the new figure 2. Make a white background and indicate the program that was used to obtain the image.

Comments on the Quality of English Language

No significant problems found

Author Response

Dear Reviewer #3:

We would like to thank you again for your constructive comments and taking the time to further review our manuscript.

Enclosed please find our revised, round two, manuscript titled "Assessment of the Performances of the Protein Modeling Techniques Participating in CASP15 Using a Structure Based Functional Site Approach: ResiRole" by Geoffrey Huang, Thomas K. Parry, and William A. McLaughlin. 

We have addressed all your comments to the best of our ability and revised the manuscript accordingly. Please see our summary and summarized responses to your comments below.

Major remark

The authors took the comments seriously and made the required changes and additions to the article, including a figure illustrating examples of predicted structures and explanations for them.

We would like to thank the reviewer again for their guidance in revising our manuscript.

Minor remark

My additional tips for the new figure 2. Make a white background and indicate the program that was used to obtain the image.

Please see that we have changed the background color for Figure 2 to white. In addition, we have added arrows to our labeling to be more specific. Lastly, we have included the source of the figure in Ln 313 (reference #28).

Sincerely yours,

William A. McLaughlin, Ph.D.
